# VAEs in the Presence of Missing Data

**Mark Collier** [1]  **Alfredo Nazabal** [2]  **Christopher K.I. Williams** [3][2]

## Abstract

Real world datasets often contain entries with missing elements e.g. in a medical dataset, a patient is unlikely to have taken all possible diagnostic tests. Variational Autoencoders (VAEs) are popular generative models often used for unsupervised learning. Despite their widespread use it is unclear how best to apply VAEs to datasets with missing data. We develop a novel latent variable model of a corruption process which generates missing data, and derive a corresponding tractable evidence lower bound (ELBO). Our model is straightforward to implement, can handle both missing completely at random (MCAR) and missing not at random (MNAR) data, scales to high dimensional inputs and gives both the VAE encoder and decoder principled access to indicator variables for whether a data element is missing or not. On the MNIST and SVHN datasets we demonstrate improved marginal log-likelihood of observed data and better missing data imputation, compared to existing approaches.

## 1. Introduction

VAEs (Kingma & Welling, 2014; Rezende et al., 2014) are a popular unsupervised learning algorithm that have found widespread application in industry (Tschannen et al., 2018). For VAEs it is typically assumed that the dataset being modelled consists of $N$ fully observed data vectors $\{\mathbf{x}^{(1)}, ..., \mathbf{x}^{(N)}\}$. Yet in practical applications this assumption is often violated (Schafer & Graham, 2002; Little & Rubin, 2019). It is common for each data vector $\mathbf{x}^{(i)}$ to consist of an observed part $\mathbf{x}_o^{(i)}$ and a missing component $\mathbf{x}_m^{(i)}$ which we do not observe. For example, relational databases often have missing columns in a row of data (Nazabal et al.,

---

[1]This work was carried out when MC was a Masters student at the University of Edinburgh. [2]The Alan Turing Institute, London, UK [3]School of Informatics, University of Edinburgh, UK. Correspondence to: Mark Collier <mcollier@tcd.ie>.

*Presented at the first Workshop on the Art of Learning with Missing Values (Artemiss) hosted by the $37^{th}$ International Conference on Machine Learning (ICML).* Copyright 2020 by the author(s).

2018) and research in the social sciences often must handle missing data (Schafer & Graham, 2002).

Existing approaches which adapt VAEs to datasets with missing data (Vedantam et al., 2017; Nazabal et al., 2018; Mattei & Frellsen, 2019; Ma et al., 2019) suffer from a number of significant disadvantages, including 1) not handling missing not at random (MNAR) data, 2) replacing missing elements with zeros with no way to distinguish an observed data element with value zero from a missing element, 3) not scaling to high dimensional inputs and/or 4) restricting the types of neural network architectures permitted, these issues are discussed in detail below. We aim to improve upon the handling of missing data by VAEs by addressing the disadvantages of the existing approaches. In particular we propose a novel latent variable probabilistic model of missing data as the result of a corruption process, and derive a tractable ELBO for our proposed model.

Our approach is straightforward to implement and has a negligible effect on the training and prediction time. We evaluate our proposed model on two computer vision datasets where we create missingness artificially by removing blocks of pixels from the image. We compare our method to (Nazabal et al., 2018) and find that our approach learns a better probabilistic model of the observed data and has lower error when imputing missing data.

## 2. Related Work

Most VAE variants assume fully observed data. Under the MCAR assumption one can integrate out the missing elements from the VAE objective and obtain an ELBO that depends only on the observed values (Nazabal et al., 2018). This corresponds to the probabilistic model in Fig. 1a (where a greyed node corresponds to an observed variable). Mattei & Frellsen (2019) take a similar approach in adapting importance weighted autoencoders (Burda et al., 2015) to missing data, assuming MAR missingness. As the encoder neural network typically expects a fixed length vector as input, the question is, what to do with the missing values in the VAE encoder input? Nazabal et al. (2018); Mattei & Frellsen (2019) choose the heuristic of replacing the missing elements with zeros. This approach has a number of issues:

1. The VAE encoder network has no way to distinguish

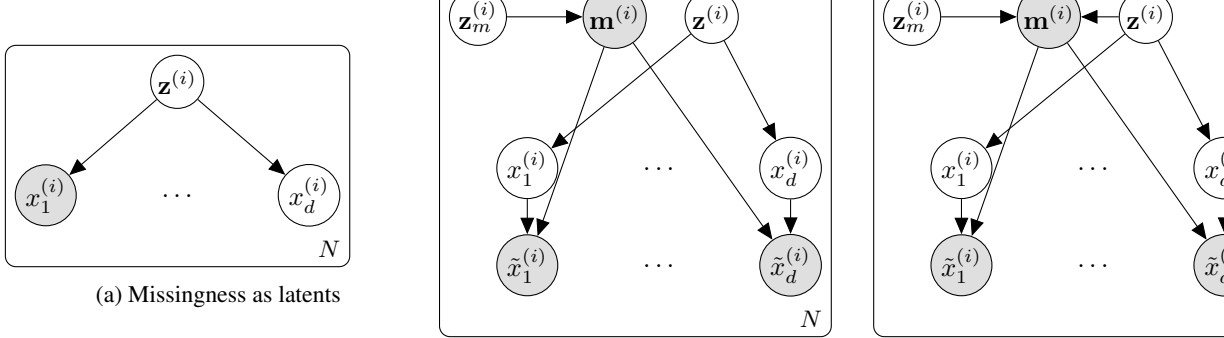

Figure 1. Latent variable models for missing data. Greyed nodes are observed, white nodes are latent.

a missing data element from an non-missing element with a value of zero.

2. It assumes MCAR/MAR missingness. In many practical applications we may observe MNAR data.

Separately it has been shown that for Factor Analysis, a simple latent variable model, that exact inference involves learning a different linear encoder for each pattern of missingness (Williams et al., 2018). The authors demonstrate the form of the exact posterior is a product of Gaussians. They speculate that a non-linear parametrization of the Factor Analysis solution may be a good inductive bias for autoencoder style neural networks.

Such an approach has in fact been implemented for VAEs under a product of Gaussians factorization for the inference model (Vedantam et al., 2017). However it requires training a separate encoder network for each dimension in $\mathbf{x}$, which does not scale to high dimensional vectors such as natural images. Vedantam et al. (2017) were primarily concerned with the ability of the VAE to generate different images based on whether certain higher level attributes were present or not, and not whether individual pixels or groups of pixels were present. Ma et al. (2019) avoid requiring a different encoder for each $\mathbf{x}$ dimension but instead require the architecture of the VAE encoder to be permutation invariant.

## 3. Proposed Model of Missing Data

We assume the existence of a data vector $\mathbf{x}^{(i)}$. Some of the components of $\mathbf{x}^{(i)}$ may be unobserved. We introduce a binary mask vector $\mathbf{m}^{(i)}$, the same length as $\mathbf{x}^{(i)}$, where $\mathbf{m}_d^{(i)} = 1$ means that $\mathbf{x}_d^{(i)}$ is observed, and 0 that it is missing.

Nazabal et al. (2018) assume the generative model in Fig. 1a where the latent variable $\mathbf{z}^{(i)}$ causes the data vector $\mathbf{x}^{(i)}$. Unobserved elements of $\mathbf{x}^{(i)}$ are integrated out of the VAE ELBO.

We propose an alternative latent variable generative model for data with missing values, which leads to a different VAE treatment addressing the issues identified with the existing approaches. Instead of treating the missing elements as latent variables we consider the observed data to be a result of a corruption process. We do not observe $\mathbf{x}^{(i)}$ but instead a corrupted version $\tilde{\mathbf{x}}^{(i)}$. We can formulate many standard imputation strategies under this process. For example for zero imputation $\tilde{\mathbf{x}}^{(i)} = \mathbf{m}^{(i)} \odot \mathbf{x}^{(i)} + (1 - \mathbf{m}^{(i)}) \odot \mathbf{0}$ and for mean imputation $\tilde{\mathbf{x}}^{(i)} = \mathbf{m}^{(i)} \odot \mathbf{x}^{(i)} + (1 - \mathbf{m}^{(i)}) \odot \boldsymbol{\mu}$ where $\boldsymbol{\mu} = \mathbb{E}[\mathbf{x}]$. We introduce an additional latent variable $\mathbf{z}_m^{(i)}$ to the standard VAE's latent $\mathbf{z}^{(i)}$, which models any structure in the missingness pattern and integrates out in the ELBO of interest to our work.

The graphical model for our proposed corruption process with MCAR missingness is shown in Fig. 1b. We note that in the MCAR case the missingness pattern $\mathbf{m}^{(i)}$ is generated *independently* of $\mathbf{z}^{(i)}$ and $\mathbf{x}^{(i)}$. In the MNAR case, there is a dependence of $\mathbf{m}^{(i)}$ on $\mathbf{z}^{(i)}$ in addition to $\mathbf{z}_m^{(i)}$, see Fig. 1c.

Under this probabilistic model we observe the missingness pattern $\mathbf{m}^{(i)}$ as well as $\tilde{\mathbf{x}}^{(i)}$, which enables us to include the missingness pattern into our VAE objective. We also note the significant advantage that our corruption process enables us to model MNAR data in a principled manner, by making the missingness pattern dependent on the latent variable $\mathbf{z}^{(i)}$ as well as $\mathbf{z}_m^{(i)}$, see Fig. 1c.

We derive a lower bound on $\log p(\mathbf{x}_o^{(i)} \mid \mathbf{m}^{(i)})$, Eq. 1, see appendix A.1 for the derivation.

$$\log p(\mathbf{x}_o \mid \mathbf{m}) \geq$$
$$\mathbb{E}_{q(\mathbf{z}|\tilde{\mathbf{x}},\mathbf{m})}\left[\log p(\mathbf{x}_o \mid \mathbf{z}, \mathbf{m})\right] - D_{KL}(q(\mathbf{z} \mid \tilde{\mathbf{x}}, \mathbf{m})||p(\mathbf{z})). \tag{1}$$

The standard VAE formulation would seek to maximize a lower bound on the log probability of all of the observed data

Original (**x**)    Mask (**m**)    Corrupted (**x̃**)    No Ind.    EO Ind.    ED Ind.

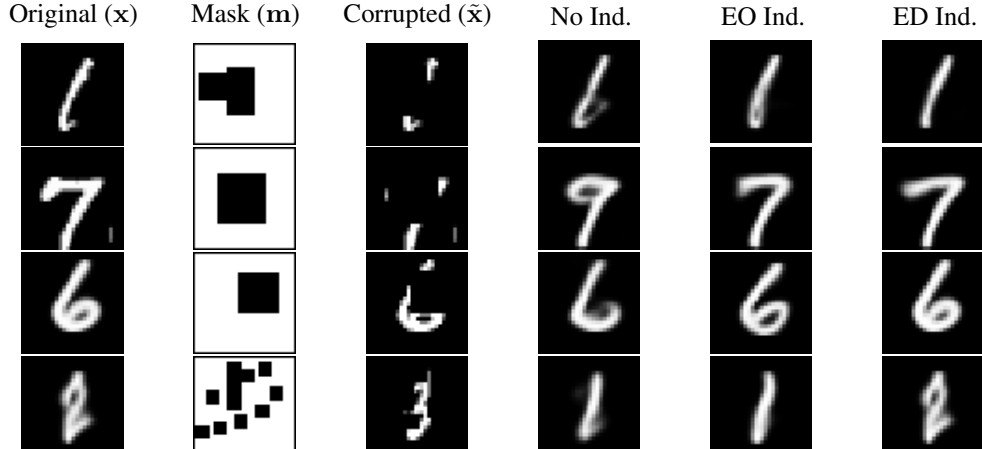

*Figure 2.* MNIST MCAR example images. Shown are the original image, missingness mask, the corrupted image and the mean reconstructions provided by the No Ind. EO Ind. and ED Ind. methods.

i.e. $\log p(\mathbf{x}_o^{(i)}, \mathbf{m}^{(i)})$. If for a particular application, one is interested in learning a generative model of the missingness pattern as well as the observed data e.g. to generate medical records with realistic missingness patterns then this would be the appropriate objective. However, we are interested in learning a good generative model of $\mathbf{x}_o$. As a result $\mathbf{m}$ is simply a modelling tool not the main object of interest. It is thus more natural to maximize $\log p(\mathbf{x}_o^{(i)} \mid \mathbf{m}^{(i)})$. We highlight that in our formulation, in addition to the VAE decoder having access to the missingness mask, the encoder $q(\mathbf{z} \mid \tilde{\mathbf{x}}, \mathbf{m})$ is also conditioned on $\mathbf{m}$. This objective is a form of conditional VAE (Sohn et al., 2015).

During training we optimize a single sample Monte Carlo estimate of Eq. 1, which under standard choices for $q(\mathbf{z} \mid \tilde{\mathbf{x}}, \mathbf{m})$, such as a multivariate Gaussian, can be computed using the reparametrization trick (Kingma & Welling, 2014; Rezende et al., 2014). We note that both the encoder $q(\mathbf{z} \mid \tilde{\mathbf{x}}, \mathbf{m})$ and the decoder $p(\mathbf{x}_o \mid \mathbf{z}, \mathbf{m})$ have access to the missingness pattern. Thus our formulation of the missing data problem for VAEs as a corruption process, and our use of a conditional VAE has enabled us to gain principled access to the missingness pattern when inferring the latent variable *and* when generating the reconstructed image, which gives the encoder and decoder an informational advantage over the existing approaches. We outline below how our model solves the issues identified previously with existing approaches:

1. The encoder network which in our approach parameterizes $q(\mathbf{z} \mid \tilde{\mathbf{x}}, \mathbf{m})$ is conditioned on $\mathbf{m}$. Thus the encoder network can distinguish between an observed data element and a missing value by looking at the corresponding element in $\mathbf{m}$.

2. As the encoder network has access to $\mathbf{m}$ its output can change based on the missingness pattern. The encoder can approximate a non-linear parameterizarion of the Factor Analysis solution of learning a separate encoder network for each missingness pattern, while sharing the same encoder weights across all missingness patterns.

3. Our model applies to both MCAR and MNAR data with no change in the training objective.

4. Our model is computationally efficient with no restrictions on the neural networks architecture used, scaling to high dimensional $\mathbf{x}$ as per standard VAEs.

## 4. Experiments

We evaluate our method on two standard computer vision datasets; the MNIST dataset of handwritten digits (LeCun et al., 2010), and the Street View House Number (SVHN) dataset (Netzer et al., 2011). We generate missing blocks of pixels randomly. Under MCAR missingness for each image we choose the number of missing blocks and each block's size from a uniform distribution over block sizes. For MNAR missingness, for each experimental run we randomly assign a fixed number and size of missing blocks to each class label e.g. all digits with label 1 will have 5 $7 \times 7$ missing blocks but digits with label 2 will have just one $15 \times 15$ missing block. Thus in the MNAR case, the number and size of the missing blocks in the image is predictive of the class label. In both the MCAR and MNAR cases once the number and size of the missing blocks is chosen, each block is positioned randomly in the image, with a uniform distribution over all positions that leave the block fully inside the image borders. See appendix B.1 for further details on the generation of missingness. Due to space constraints we leave network architecture and optimization details to appendices B.2 and B.3 respectively[1].

---

[1] Code to reproduce our results: https://github.com/MarkPKCollier/MissingDataVAEs.

*Table 1.* MCAR/MNAR results. Each cell contains two values the metric for MCAR missingness and the metric for MNAR missingness. For each metric we conduct a paired sample t-test between the method and ED Ind. * indicates p < 0.001. 60 replicates for the t-test are paired by generating a new dataset as per the process outlined in appendix B.1 followed by training and evaluation of each method.

| METHOD | MNIST | | SVHN | |
|---|---|---|---|---|
| | $\log p(\mathbf{x}_o \mid \mathbf{m})$ | $\mathbb{E}_q\left[\log p(\mathbf{x}_m \mid \mathbf{z}, \mathbf{m})\right]$ | $\log p(\mathbf{x}_o \mid \mathbf{m})$ | $\mathbb{E}_q\left[\log p(\mathbf{x}_m \mid \mathbf{z}, \mathbf{m})\right]$ |
| NO IND. | $-63.95^*/-64.21^*$ | $-64.03^*/-65.66^*$ | $-5934.42^*/-5967.43^*$ | $-3065.70^*/-3074.08^*$ |
| EO IND. | $-63.04/-63.25$ | $-58.28^*/-59.09^*$ | $-5853.79/-5863.37$ | $-3175.00^*/-3198.89^*$ |
| ED IND. | $\mathbf{-62.99/-63.23}$ | $\mathbf{-56.11/-57.01}$ | $\mathbf{-5828.02/-5841.13}$ | $\mathbf{-2844.40/-2944.98}$ |

The methods we compare can be distinguished based on whether the VAE encoder or decoder have access to $\mathbf{m}$. In (Nazabal et al., 2018) neither the encoder or decoder has access to the missing indicators, so we call this method No Ind. Our primary method gives both the encoder and decoder access to $\mathbf{m}$, we call our method ED (Encoder Decoder) Ind. For the purposes of an ablation study on the effect of conditioning the decoder on $\mathbf{m}$ we introduce a variant of our method, EO (Encoder Only) Ind. where only the encoder has access to $\mathbf{m}$.

We are interested in how the methods compare in two ways:

1. **Modelling the observed data**. Better models should assign higher log-likelihood to the observed data. As is standard in the VAE literature (Kingma et al., 2016), to evaluate this criterion, we estimate the test set marginal log-likelihood $\log p(\mathbf{x}_o \mid \mathbf{m})$ using importance sampling (with 256 importance samples).

2. **Imputing missing data**. Our method maximizes a lower bound on the log-likelihood of the observed data. This objective does not explicitly encourage missing data imputation. However, we expect an efficient decoding scheme to make good predictions for all variables and hence provide useful imputations. To measure this, we report a MC estimate (256 samples) of $\mathbb{E}_{q(\mathbf{z} \mid \tilde{\mathbf{x}}, \mathbf{m})}\left[\log p(\mathbf{x}_m \mid \mathbf{z}, \mathbf{m})\right]$ on the test set.

In Table 1 we report an importance sampled estimate of the marginal log-likelihood and our imputation quality metric. Metrics are evaluated on the test set for both datasets in MCAR and MNAR settings. Paired t-tests with 60 replicates matched by corresponding random seeds are used to evaluate the significance of any differences in these metrics. Further metrics, including the bits per pixel metric and mean squared error are provided in appendix C.1.

We see that for both MNIST and SVHN in the MCAR and MNAR settings the variants of our method; EO Ind. and ED Ind. model the observed data better and provide better imputation of the missing data than the approach from the literature, No Ind. As anticipated, providing the decoder with the missing mask $\mathbf{m}$ improves performance over the EO Ind. method in which only the encoder has access to $\mathbf{m}$. However the effect sizes are small and not statistically significant for the marginal log-likelihood of the observed data when comparing the EO and ED method.

We present some example test set images from MNIST under MCAR missingness and mean reconstructions for each method in Fig. 2. Similar visualizations are provided for MNIST with MNAR missingness and SVHN under both MCAR and MNAR missingness in appendix C.2. The visualizations give a sense of why the methods which have access to $\mathbf{m}$ outperform the No Ind. method on the quantitative metrics. From the corrupted image $\tilde{\mathbf{x}}$ of the 7 in row 2 of Fig. 2, it is ambiguous as to whether the original image was a 9 or a 7, however the missingness mask disambiguates this to an extent. Likewise in the last row we see a heavily corrupted 2, both the No Ind. and EO Ind. methods reconstruct what appears to be a 1, however by providing the decoder access to the rather complex missing mask $\mathbf{m}$ in the ED Ind. method we get a very good reconstruction.

## 5. Conclusion

We have addressed the question of how to apply VAEs to high-dimensional datasets with missing values. Previous approaches in the literature have made restrictive assumptions on the type of missingness, encoder network architecture and imputation strategy. We have proposed a novel generative model of missing data as a corruption process. We model the observed data as a conditional VAE and have derived the ELBO for $\log p(\mathbf{x}_o \mid \mathbf{m})$ under this model. Both MCAR and MNAR cases can be handled elegantly. Our method gives the encoder $q(\mathbf{z} \mid \tilde{\mathbf{x}}, \mathbf{m})$ and the decoder principled access to the missingness pattern $\mathbf{m}$ with no restrictions on the network architectures used.

Empirically, our method learns a better probabilistic model of the observed data and provides improved missing data imputation than the baseline method from the literature. We believe our work provides a simple yet effective method for applying VAEs to datasets with missing elements, and that our latent variable generative model of missing data will prove useful in adapting other popular latent variable probabilistic models to datasets with missing data.

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

# A. ELBO derivations

## A.1. ELBO derivation for $\log p(\mathbf{x}_o^{(i)} \mid \mathbf{m}^{(i)})$

We set our prior $p(\mathbf{z})$ to be independent of $\mathbf{m}$ i.e. $p(\mathbf{z} \mid \mathbf{m}) = p(\mathbf{z})$.

$$\log p(\mathbf{x}_o \mid \mathbf{m}) = \log \int \int p(\mathbf{x} \mid \mathbf{z}, \mathbf{m}) p(\mathbf{z}) d\mathbf{x}_m d\mathbf{z}$$

$$= \log \int p(\mathbf{x}_o \mid \mathbf{z}, \mathbf{m}) p(\mathbf{z}) d\mathbf{z}$$

$$= \log \int p(\mathbf{x}_o \mid \mathbf{z}, \mathbf{m}) p(\mathbf{z}) \frac{q(\mathbf{z} \mid \tilde{\mathbf{x}}, \mathbf{m})}{q(\mathbf{z} \mid \tilde{\mathbf{x}}, \mathbf{m})} d\mathbf{z}$$

$$= \log \mathbb{E}_{q(\mathbf{z} \mid \tilde{\mathbf{x}}, \mathbf{m})} \left[ p(\mathbf{x}_o \mid \mathbf{z}, \mathbf{m}) \frac{p(\mathbf{z})}{q(\mathbf{z} \mid \tilde{\mathbf{x}}, \mathbf{m})} \right]$$

$$\geq \mathbb{E}_{q(\mathbf{z} \mid \tilde{\mathbf{x}}, \mathbf{m})} \left[ \log p(\mathbf{x}_o \mid \mathbf{z}, \mathbf{m}) \right] - D_{KL}(q(\mathbf{z} \mid \tilde{\mathbf{x}}, \mathbf{m}) || p(\mathbf{z}))$$

# B. Experimental Setup

## B.1. Datasets

We use two standard computer vision datasets; 1) the MNIST dataset of handwritten digits (LeCun et al., 2010), and 2) the Street View House Number (SVHN) dataset (Netzer et al., 2011). MNIST consists of 70,000 greyscale 28x28 images of handwritten numbers 0-9. We use the standard train/test split of 60,000 training images and 10,000 test set images. Of the 60,000 training images we use 10,000 as a validation set. We use 10,000 of the 73,257 labelled training images in SVHN dataset as a validation set with the remaining 63,257 used for training and the standard 26,032 test images. The SVHN dataset consists of 32x32 RGB images of house numbers captured from Google Street View which may have multiple numbers in view but are cropped such that a single digit from 0-9 is most prominent. We scale each element of both datasets into $[0, 1]$ by dividing elementwise by 255.

Despite the pixels in the MNIST dataset being real valued values in $[0, 1]$ we use a Bernoulli likelihood model as is common in the VAE literature (Loaiza-Ganem & Cunningham, 2019). This can be justified by considering each pixel to be a probability of emission of a black or white pixel[2]. Such an interpretation does not reasonably extend to the RGB SVHN dataset, despite some attempts to do just that (Gregor et al., 2015). Thus following (Salimans et al., 2017) we choose to model each RGB component of each SVHN pixel using the Logistic distribution. The Logistic distribution has a mean and scale parameter similar to a Gaussian distribution however has heavier tails and its cumulative distribution function (CDF) is the sigmoid function. The Logistic is used for precisely these reasons as the heavier tails makes our model less sensitive to outliers than the Gaussian and access to an analytically computable CDF

enables computation of the log density at the boundaries where pixel components take on the values of 0 or 255. We refer the reader to (Salimans et al., 2017) for a fuller discussion of the Logistic distribution for modelling RGB pixels and we note that we make use of the authors' numerically stable Tensorflow implementation of the logistic mixture distribution[3].

We generate missingness in our otherwise fully observed datasets artificially. This enables us to measure the models' performance at both reconstruction of observed data and imputation of missing data. We wish to test our model on both MCAR and MNAR data.

We assign the block size and number of blocks randomly for each experimental run. We generate several blocks of missing pixels for each image. For MNAR missingness, we vary the number and size of the missingness blocks depending on the class label for the image. Whereas for MCAR missingness, the number and size of the missing blocks is drawn at random independently for each image, with equal probability on all block sizes.

For MNIST an image (MCAR) or digit class (MNAR) could be assigned between 10 $5 \times 5$ missing blocks up to a single $15 \times 15$ missing block. While for SVHN this ranged from 12 $5 \times 5$ missing blocks up to one $17 \times 17$ block. For MNIST this resulted in 23% of the pixels being dropped out and 20% for SVHN. In particular for MNIST each image (MCAR)/each of the 10 MNIST classes (MNAR) is randomly assigned 10 $5 \times 5$, 12 $5 \times 5$, 5 $7 \times 7$, 6 $7 \times 7$, 3 $9 \times 9$, 4 $9 \times 9$, 2 $11 \times 11$, 3 $11 \times 11$, 1 $13 \times 13$ or 1 $15 \times 15$ missing blocks. Then for each image (MCAR)/image from that class (MNAR) the corresponding number and size of missing blocks are removed from the image. For example if the digit 2 for the MNAR MNIST dataset is randomly assigned 5 $7 \times 7$ blocks for a particular experimental run,

---

then for every digit 2 in the dataset we place 5 $7 \times 7$ missing blocks centered at randomly selected pixels such that the blocks may overlap but may not extend beyond the image boundary. For SVHN the generation process is the same, but the possible number of sizes of the missing blocks are: 12 $5 \times 5$, 5 $7 \times 7$, 6 $7 \times 7$, 3 $9 \times 9$, 4 $9 \times 9$, 2 $11 \times 11$, 3 $11 \times 11$, 2 $13 \times 13$, 1 $15 \times 15$ or 1 $17 \times 17$ missing blocks.

By exploiting the dependence of the missingness pattern on the class labels in the MNAR case, the models should be better able to infer the latent variable $\mathbf{z}$ and thus impute missing data and reconstruct observed data.

## B.2. Network Architectures

For the MNIST dataset we use an encoder network architecture loosely inspired by the LeNet-5 (LeCun et al., 1998). The dimensionality of our latent variable $\mathbf{z}$ is chosen to be 50. The encoder consists of two convolutional layers, the first with 20 5x5 filters, a stride of 2 and ReLU activation followed by 40 5x5 filters, a stride of 2 and ReLU activation. A fully-connected linear layer then outputs the mean parameter of $q(\mathbf{z}|\mathbf{x})$ and a fully-connected layer with softplus activation outputs the standard deviation of $q(\mathbf{z}|\mathbf{x})$. The decoder architecture initially mirrors the encoder architecture with a fully connected layer with $7 * 7 * 20$ output units and ReLU activation being followed by 2 transposed convolutional layers with 40 5x5 and 20 5x5 filters, each with a stride of 2 and ReLU activation. If the decoder has access to $\mathbf{m}$ it is now concatenated to the output of the transposed convolutions. In either case a further three standard convolutional layers follow with 10 5x5, 10 5x5 and 1 3x3 filters respectively, all with stride 1, the first two with ReLU activation and the last with sigmoid activation. For all convolutional and transposed convolution layers "SAME" padding is used (Dumoulin & Visin, 2016).

The encoder and decoder networks for the SVHN dataset have the same form but vary in size from the MNIST architecture. In particular, the SVHN latent variable is 200 dimensional. The encoder consists of three convolutional layers of 40 3x3, 60 3x3 and 60 5x5 filters, each with a stride of 2 and ReLU activation followed by a fully-connected layer outputting the parameters of $q(\mathbf{z}|\mathbf{x})$. The decoder network has a fully-connected layer with $4 * 4 * 60$ output units and ReLU activation followed by 3 transposed convolutional layers to mirror the encoder convolutional layers. Again if the decoder has access to $\mathbf{m}$ it is now concatenated to the hidden layer. Three standard convolutional layers with 30 5x5, 30 5x5 and 2 3x3 filters follow, all with stride of 1, the first two with ReLU activation and the last with no activation for the outputs corresponding to the mean parameter of the logistic distribution and exponential activation for the scale parameter. Again "SAME" padding is used for all convolutional and transposed convolutional layers.

## B.3. Optimization Details

We use the Adam optimizer (Kingma & Ba, 2014) with initial learning rate of $10^{-3}$ and a batch size of 256. We train with early stopping for a maximum of 200 epochs, stopping if the MC estimate of the validation set ELBO has not improved in 10 epochs. We add $10^{-3}$ to each dimension of the encoder output defining the posterior standard deviation on $\mathbf{z}$, ensuring the posterior variance is never below $10^{-6}$. This avoids a common optimization difficulty with VAEs where the posterior variance is driven to zero. Likewise the scale parameter for the logistic mixture distribution used for RGB images is lower bounded by $10^{-2}$.

# C. Additional Experimental Results

## C.1. Additional Metrics

*Table 2.* MNIST MCAR/MNAR additional results. Each cell contains two values the metric for MCAR missingness and the metric for MNAR missingness. For each metric we conduct a paired sample t-test between the method and ED Ind. * indicates p < 0.001, 60 replicates are used for the t-test. $\hat{\mathbf{x}}_o = \mathbb{E}_q\left[p(\mathbf{x}_o \mid \mathbf{z}, \mathbf{m})\right]$ and similarly $\hat{\mathbf{x}}_m = \mathbb{E}_q\left[p(\mathbf{x}_m \mid \mathbf{z}, \mathbf{m})\right]$. Mean square error metrics $(\mathbf{x}_o - \hat{\mathbf{x}}_o)^2$ are shown on a per pixel level.

| METHOD | BITS/PIXEL | $(\mathbf{x}_o - \hat{\mathbf{x}}_o)^2$ | $(\mathbf{x}_m - \hat{\mathbf{x}}_m)^2$ |
|---|---|---|---|
| NO IND. | .1519*/.1526* | .0118*/.0121* | .0696*/.0716* |
| EO IND. | .1497/.1503 | .0111*/.0114* | .0645*/.0658* |
| ED IND. | **.1496/.1502** | **.0110/.0112** | **.0635/.0650** |

*Table 3.* SVHN MCAR/MNAR additional results. Each cell contains two values the metric for MCAR missingness and the metric for MNAR missingness. For each metric we conduct a paired sample t-test between the method and ED Ind. * indicates p < 0.001. 60 replicates are used for the t-test. $\hat{\mathbf{x}}_o = \mathbb{E}_q\left[p(\mathbf{x}_o \mid \mathbf{z}, \mathbf{m})\right]$ and similarly $\hat{\mathbf{x}}_m = \mathbb{E}_q\left[p(\mathbf{x}_m \mid \mathbf{z}, \mathbf{m})\right]$. Mean square error metrics $(\mathbf{x}_o - \hat{\mathbf{x}}_o)^2$ are shown on a per pixel level.

| METHOD | BITS/PIXEL | $(\mathbf{x}_o - \hat{\mathbf{x}}_o)^2$ | $(\mathbf{x}_m - \hat{\mathbf{x}}_m)^2$ |
|---|---|---|---|
| NO IND. | 10.4750*/10.5363* | .0006*/.0006* | **.0084*/.0086*** |
| EO IND. | 10.3328/10.3527 | .0005/.0005 | .0086*/.0087* |
| ED IND. | **10.2870/10.3134** | **.0005/.0005** | .0101/.0103 |

For SVHN the ED Ind. method provides the best imputation when measured by the expected log probability of the missing data but when imputation quality is measured by mean squared error ED Ind. has the worst imputation. We attribute this to the mismatch between training and evaluation metric.

## C.2. Visualizations

Original (**x**)   Mask (**m**)   Corrupted (**x̃**)   No Ind.   EO Ind.   ED Ind.

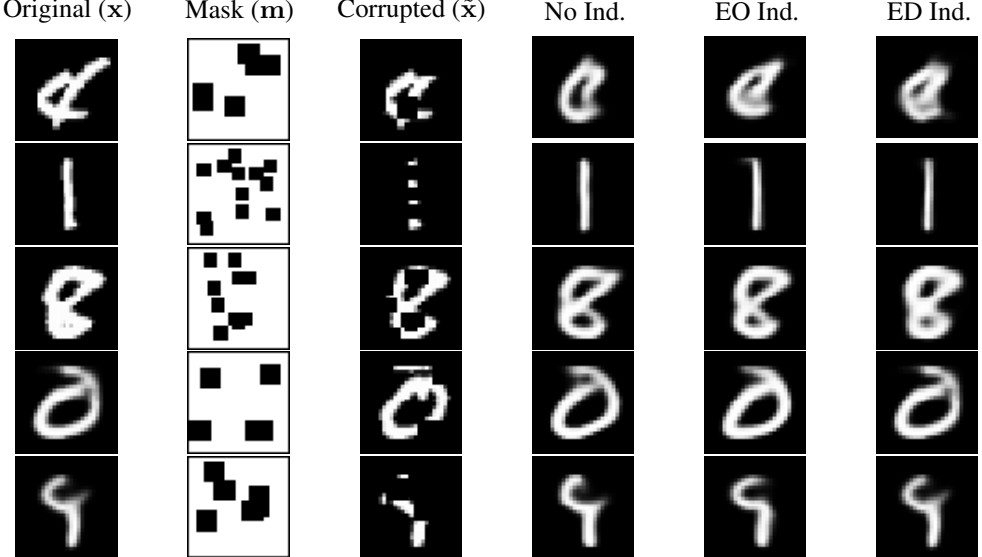

*Figure 3.* MNIST MNAR example images. Shown are the original image, missingness mask, the corrupted image and the mean reconstructions provided by the No Ind. EO Ind. and ED Ind. methods.

Original (**x**)   Mask (**m**)   Corrupted (**x̃**)   No Ind.   EO Ind.   ED Ind.

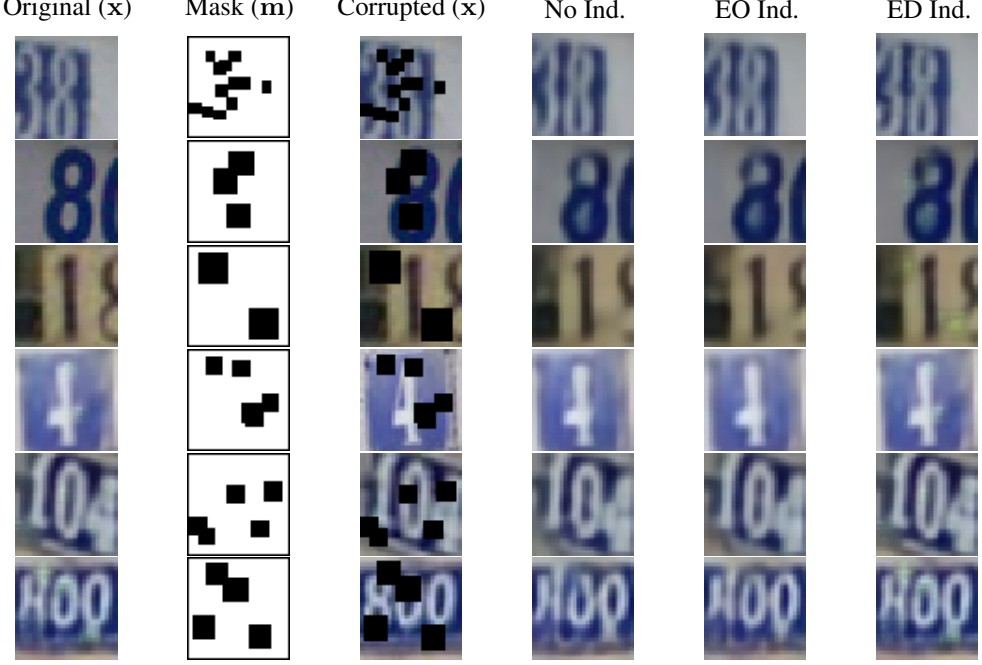

*Figure 4.* SVHN MCAR example images. Shown are the original image, missingness mask, the corrupted image and the mean reconstructions provided by the No Ind. EO Ind. and ED Ind. methods.

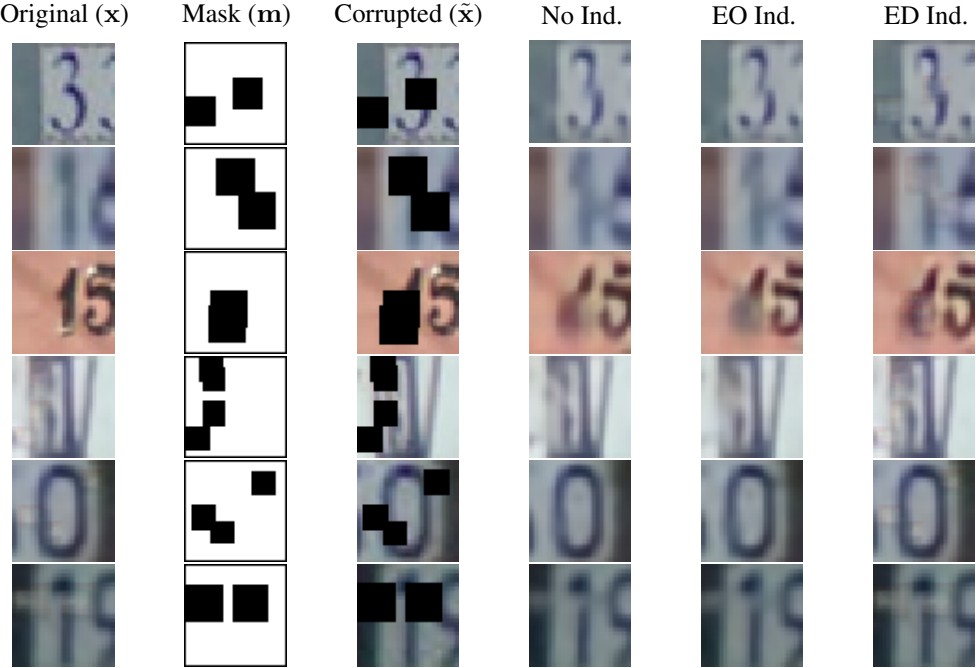

*Figure 5.* SVHN MNAR example images. Shown are the original image, missingness mask, the corrupted image and the mean reconstructions provided by the No Ind. EO Ind. and ED Ind. methods.