# OpenReview forum: "VAEs in the Presence of Missing Data"
_ICML.cc/2020/Workshop/Artemiss — ICML Artemiss 2020_

### Official Review · AnonReviewer1 · 2020-06-23
**Review of VAEs in the Presence of Missing Data**

**Confidence:** 3
**Rating:** 7

**Review:**

This paper addresses the issue of handling missing data with VAEs. The proposed model should be able to break away from the M(C)AR assumption. The missing processes is formulated as a corruption process and a conditional VAE, conditional on the mask, is used to model more general missing processes than MCAR. Experiments on image datasets show model performance, compared to the HI-VAE.

The topic of missing data in VAEs is relevant and the idea is good.

Comments:
- There is an emerging literature on this topic, with notable contributions such as [1] and [2], where [2] has been compared directly to the HI-VAE. These two approaches have come up with different strategies for avoiding 0-imputation or the effects of this, and both scale well to high-dimensional inputs.
- Could you make it clear how the objective changes when going from the MCAR assumption to the MNAR assumption?
- Is it really more natural to model p(x|m) than p(x,m) in an MNAR setting? it looks like you are actually implcitily doing the later, taking a pattern mixture model decomposition of p(x,m)=p(x|m)p(m), and then giving each missing pattern equal weight.
- A comment on why the proposed model has the worst imputation error on SVHN would be nice.

[1] Ma, Chao, et al. "Eddi: Efficient dynamic discovery of high-value information with partial vae." arXiv preprint arXiv:1809.11142 (2018).

[2] Mattei, Pierre-Alexandre, and Jes Frellsen. "Miwae: Deep generative modelling and imputation of incomplete data." arXiv preprint arXiv:1812.02633 (2018).

---

### Decision · Program_Chairs · 2020-07-02

**Decision:**

Accept

**Comment:**

We're happy to accept this paper at Artemiss. We'll contact you soon to inform you about more details concerning the format of your presentation at the workshop, and the camera-ready version deadline. Please take into account the referee's comments to write the camera-ready version.